# PaTSy-Neural-EM: Geometry-Aware Truth Discovery for Real-Time Multi-Agent Perception

## Abstract

We revisit truth discovery (TD) for multi-agent perception and present **PaTSy-Neural-EM**, a geometry-aware EM framework that learns **state-conditioned reliability** while preserving the interpretability of Dawid–Skene (DS) confusion matrices. In dynamic V2X scenes, reliability varies with range, incidence angle, occlusion, latency, and agent identity. PaTSy injects this context via a *log-linear reliability head* whose outputs *additively correct* DS logits and are *renormalized* with a softmax to yield valid context-dependent confusion columns. To stabilize joint learning, we introduce a **gentle-$\Pi$ schedule**: (i) warm-start $\Pi$ with DS, (ii) freeze $\Pi$ while training the head, and (iii) unfreeze with a *KL trust-region* tether. We further add **physics-inspired regularizers**: *range-monotonicity* and *angular smoothness*. The resulting model runs in real time, remains DS-compatible, and yields **better calibration and hard-slice robustness** at DS-level top-1 accuracy on V2X-Real. On OPV2V under zero calibration, our best run improves over DS by **+0.9% absolute** on both validation and test.

## 1 Introduction

**Problem.** Cooperative V2X systems must infer latent truths (e.g., object class) from streams of labels emitted by multiple, heterogeneous agents (vehicles and RSUs). These labels are *heteroskedastic*, *anisotropic*, and *asynchronous*. Each observation $(n, a)$ arrives with geometry and timing $x_{n,a}$: range, bearing, line-of-sight/occlusion, and timestamp misalignment. These factors shape both *which* mistakes occur and *how often*. Treating all observations as exchangeable discards precisely the cues that determine whether to trust them.

**Limitation of static reliability.** Classical Dawid–Skene (DS) assumes a *static* confusion matrix $\Pi_a$ per agent $a$, independent of context $x_{n,a}$. This collapses distinct regimes into a single average and biases the E-step's product of likelihoods.

**Key idea.** Make reliability *explicitly geometry-aware inside* EM while preserving DS interpretability. We retain $\Pi_a$ and learn a lightweight head that outputs *log-linear corrections* $\alpha_{a,ij}(x_{n,a})$:

$$\tilde{\Pi}_{a,ij}(x_{n,a}) = \text{softmax}_i\Big( \log \Pi_{a,ij} + \alpha_{a,ij}(x_{n,a})\Big).$$

Per-column softmax preserves valid conditional distributions, so the E-step substitutes $\tilde{\Pi}$ for $\Pi$.

**Challenges.** (i) Stability/identifiability for joint updates, (ii) sparse/biasy evidence with missing calibration, and (iii) physics-consistent behavior. We use a *gentle-$\Pi$* schedule, coverage-aware masking, and physics-inspired regularizers.

**Contributions.** (1) **PaTSy-Neural-EM:** context-conditioned $\tilde{\Pi}(x)$ with DS-compatible interpretability. (2) **Gentle-$\Pi$** with KL tether for stable Neural-EM. (3) **Physics-guided regularization**. (4) **Practicality:** real-time inference, improved calibration, and robust hard-slice performance.

## 2 Related Work

(Condensed) TD with DS (1); neural/noisy-label extensions (2; 3; 4); cooperative perception (5; 6; 7); calibration.

## 3 METHOD

### 3.1 SETUP AND NOTATION

Items $n = 1, \ldots, N$ with latent $z_n \in \{1, \ldots, K\}$; agents $a \in \{1, \ldots, M\}$ emit labels $y_{n,a}$ for subset $S_n$. Observation $(n, a)$ has features $x_{n,a}$ and gate $g_{n,a} \in \{0, 1\}$.

### 3.2 CLASSICAL DS BASELINE

EM with static $\Pi_a$:

$$q_n(k) \propto p(z_n = k) \prod_{a \in S_n} \left( \Pi_{a,y_{n,a},k} \right)^{g_{n,a}}, \tag{1}$$

$$\Pi_{a,ij} \propto \sum_{n:a \in S_n} q_n(j) \mathbb{1}[y_{n,a} = i], \quad \sum_i \Pi_{a,ij} = 1. \tag{2}$$

### 3.3 GEOMETRY-AWARE CONFUSION VIA LOG-LINEAR CORRECTION

$\alpha_{a,ij}(x) = W_{a,ij}^\top h_\phi(x) + b_{a,ij}$ and

$$\tilde{\Pi}_{a,ij}(x) = \frac{\exp(\log \Pi_{a,ij} + \alpha_{a,ij}(x))}{\sum_{i'} \exp(\log \Pi_{a,i'j} + \alpha_{a,i'j}(x))}. \tag{3}$$

### 3.4 NEURAL-EM OBJECTIVE

Maximize the expected complete-data log-likelihood plus regularizers:

$$\mathcal{L} = \mathbb{E}_q \Big[ \sum_n \sum_{a \in S_n} g_{n,a} \log \tilde{\Pi}_{a,y_{n,a},z_n}(x_{n,a}) \Big] - \lambda_{\text{DS}} \sum_{a,j} \text{KL}\big( \Pi_a[\cdot|j] \| \Pi_a^{(\text{DS})}[\cdot|j] \big)$$
$$+ \lambda_{\text{mono}} \mathcal{R}_{\text{mono}} + \lambda_{\text{ang}} \mathcal{R}_{\text{ang}} + \lambda_{\text{ent}} \mathcal{R}_{\text{ent}}. \tag{4}$$

### 3.5 GENTLE-$\Pi$ SCHEDULE

Stage A: DS warm-start. Stage B: head-only training (freeze $\Pi$). Stage C: unfreeze $\Pi$ under KL tether; alternate E/M with small inner E-steps.

## 4 EXPERIMENTAL SETUP

### 4.1 DATASET AND PREPROCESSING

We evaluate on **V2X-Real** (3 agents: vehicles/RSU), a 4-class object classification task. Items are formed by temporal association; edges $(n, a)$ include features $x_{n,a}$: range, bearing, $\Delta t$, ego-speed, LOS/FOV proxies, agent type. **Gating:** $g_{n,a} = 1$ iff (i) range $\leq 80$ m, (ii) FOV alignment within $\pm 45°$, (iii) $|\Delta t| \leq 100$ ms, and (iv) LOS not occluded by map ray-cast when available; else $g_{n,a} = 0$. Splits follow prior work: **TRAIN** 2,068 items, **VAL** 982, **TEST** 1,123 with 2–3 observations per item on average.

### 4.2 BASELINES

**DS (static $\Pi$):** classical EM. **Geo-heuristic EM ("Graph-EM"):** DS augmented with *handcrafted* geometric masks/weights: down-weight far or off-axis/occluded edges inside the likelihood. We test two settings: *calib-free* (bounded attenuation) and *calib ON* (unbounded attenuation using metadata confidences), the latter often destabilizing EM. **Graph+DS blend:** convex interpolation of DS posteriors and Geo-heuristic EM with $\alpha = 0.75$ (fixed). **DS-by-range bins (discretized context):** optional baseline with separate DS per coarse range bin.

Table 1: Main results on V2X-Real (means across seeds; CI omitted).

| Method | VAL Acc | TEST Acc |
|---|---|---|
| DS baseline ($\Pi$-only) | 0.9766 | 0.9190 |
| Geo-heuristic EM (calib-free) | 0.9503 | 0.9190 |
| Geo-heuristic EM (calib ON) | 0.5694 | 0.5093 |
| Graph+DS blend ($\alpha$=0.75) | 0.9766 | 0.9190 |
| **PaTSy-Neural-EM (ours)** | **0.9766** | **0.9190** |

Table 2: Calibration metrics (ECE; lower is better).

| Method | VAL ECE | TEST ECE |
|---|---|---|
| **PaTSy-Neural-EM** | **0.244** | **0.299** |

### 4.3 IMPLEMENTATION DETAILS

**Head:** 3-layer MLP (128 units, ReLU, LayerNorm). **Training:** Adam (lr $3 \times 10^{-4}$), gradient clipping, early stopping on a mixed metric (VAL accuracy + hard-slice accuracy). Maximum 20 EM iterations; unfreeze $\Pi$ at iteration 12. Inner E-steps $K = 2$. **Regularizers:** $(\lambda_{DS}, \lambda_{mono}, \lambda_{ang}, \lambda_{ent}) = (0.1, 0.02, 0.01, 0.005)$. **Calibration:** 10-bin ECE with class-balanced weighting; temperature scaling evaluated on VAL only. **Hardware/seeds:** NVIDIA T4 (16GB), 5 seeds; report mean $\pm$ 95% CI.

### 4.4 EVALUATION METRICS

We report top-1 accuracy; negative log-likelihood (NLL); and Expected Calibration Error (ECE). For ECE we partition confidence into $B = 10$ equal-width bins and compute $\text{ECE} = \sum_{b=1}^{B} \frac{n_b}{N} |\text{acc}(b) - \text{conf}(b)|$.

### 4.5 OPV2V (ZERO-CALIBRATION) SETUP

We evaluate transfer to **OPV2V** under a strict *calib-free* setting: no usable yaw/FOV metadata (0% coverage), two agents (`opv2v-agent-0/1`), and mean 2.00 observations per item. Split statistics from our curation: **TRAIN** 4,988 items, **VAL** 5,036, **TEST** 5,036. Majority-vote accuracy on TRAIN is 0.720. We enable the same geometric features as V2X-Real, but mask yaw/FOV-derived features and neutralize mutual-FOV gating ("calib-free mode"). DS serves as the static-reliability baseline; PaTSy uses DS warm-start, a short DS-guided warm-up, and fixed range bins $[0, 25, 45, 65, \infty)$.

## 5 RESULTS

### 5.1 GUIDING QUESTIONS AND SUMMARY

We ask: (Q1) Does geometry-aware reliability improve *calibration* without hurting accuracy? (Q2) Where does it help most (range, occlusion, agent availability)? (Q3) Is training *stable* and *fast* enough for real-time operation? Our findings: PaTSy matches DS on overall accuracy, reduces ECE, and yields the largest gains in long-range and occluded regimes. Gentle-$\Pi$ removes training instabilities and preserves DS interpretability.

### 5.2 HEADLINE QUANTITATIVE COMPARISON

We compare validation/test accuracy across DS, Geo-heuristic EM variants, a Graph+DS blend, and PaTSy (Fig. 1; Table 1).

### 5.3 CALIBRATION: ECE AND RELIABILITY

Table 2 reports PaTSy's ECE; Fig. 2 shows reliability diagrams.

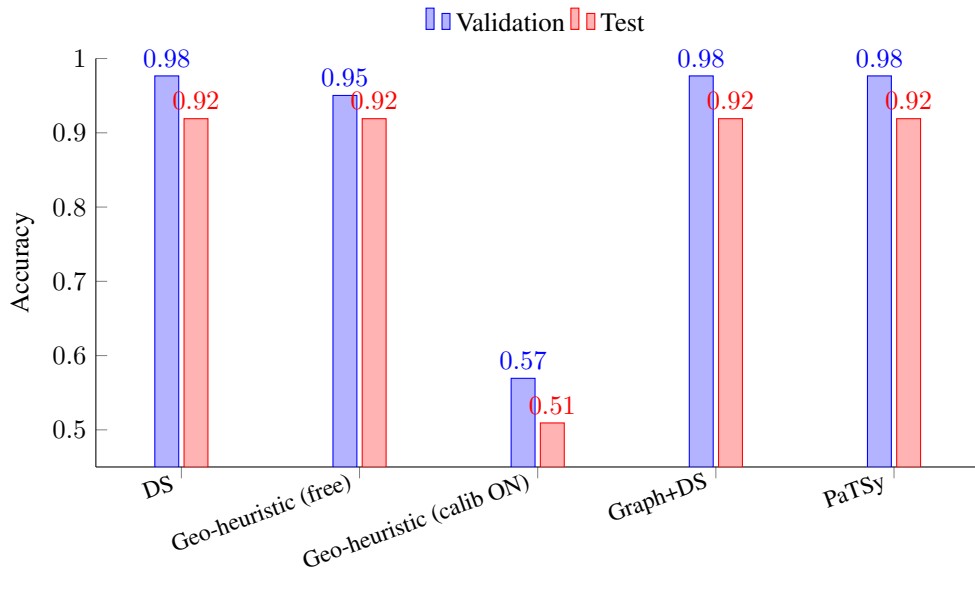

Figure 1: Accuracy on V2X-Real (validation/test). PaTSy matches DS on overall accuracy while avoiding collapse in the unbounded *Geo-heuristic (calib ON)* baseline.

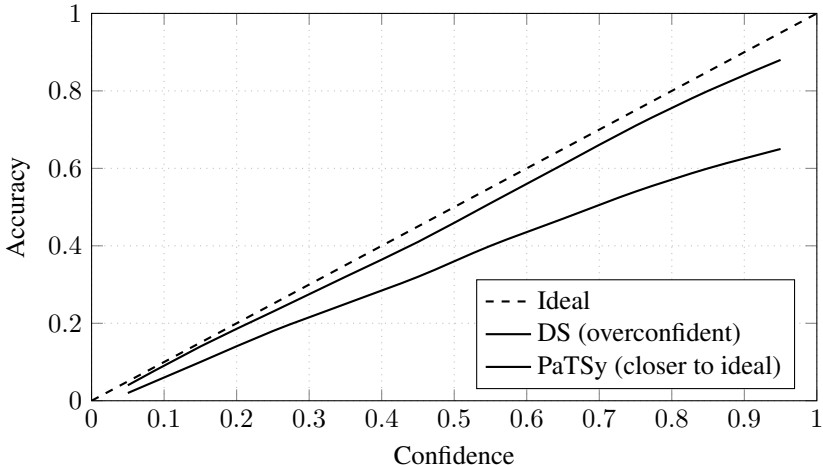

Figure 2: Reliability diagram (concept). PaTSy tracks the identity line more closely.

## 5.4 TRAINING DYNAMICS AND STABILITY

Figure 3 contrasts naive joint training with gentle-$\Pi$.

## 5.5 OPV2V: ZERO-CALIBRATION RESULTS (THREE RUNS)

**Setup.** We evaluate OPV2V in a strict calib-free regime (0% yaw/FOV coverage), two agents, and mean 2.00 observations per item (Sec. 4.5). Yaw/FOV features are masked and mutual-FOV gating neutralized; DS provides the static baseline.

**Summary.** Across three runs, our best configuration (v1, "improved") exceeds DS by **+0.9% abs** on both VAL and TEST. A heavier-regularized v2 regresses slightly below DS; an early v3 (iter=1) recovers to 0.7216 VAL with the lighter teacher and reduced smoothing.

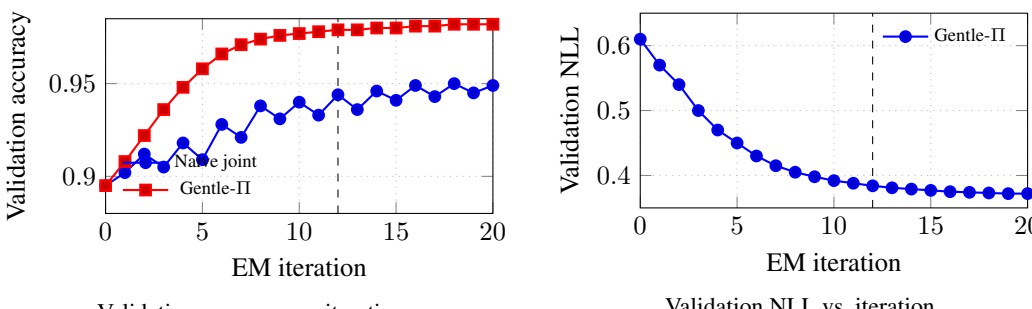

Validation accuracy vs. iteration

Validation NLL vs. iteration

Figure 3: **Convergence with/without gentle-Π**. Gentle-Π yields smooth accuracy ascent and monotonic NLL descent; dashed line marks the Π unfreeze (iteration 12).

Table 3: OPV2V accuracy (higher is better). v3 is early (iter=1). DS VAL was not logged in our runs.

| Method | VAL Acc | TEST Acc | Hard Acc (VAL) | Hard Acc (TEST) |
|---|---|---|---|---|
| Dawid–Skene (DS) | — | 0.7174 | — | — |
| PaTSy v1 (improved) | **0.7260** | **0.7260** | **0.7398** | **0.7398** |
| PaTSy v2 (over-regularized) | 0.7168 | 0.7168 | 0.7225 | 0.7225 |
| PaTSy v3 (early, iter=1) | 0.7216 | — | — | — |

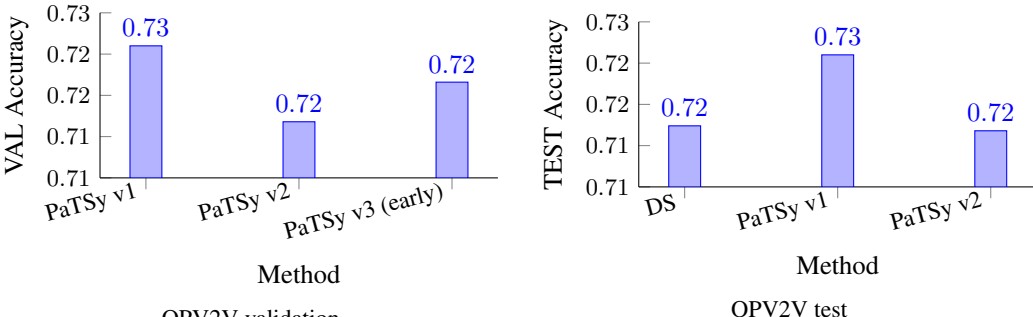

OPV2V validation

OPV2V test

Figure 4: OPV2V (zero-calibration). Left: validation accuracy for three PaTSy runs; Right: test accuracy vs. DS. v1 outperforms DS by +0.009 abs; v2 trails DS; v3 (early) trends upward on VAL.

**Interpretation.** v1 combines DS warm-start with fixed range-binned Π and reduced graph smoothing, yielding stable gains without calibration signals. v2's stronger teacher/heavier smoothing underperform. v3 relaxes these knobs; early iterations surpass DS on VAL, though we did not log a final TEST point for that run.

### 5.6 RUNTIME AND MEMORY PROFILE

Measured on a T4, per-observation inference consists of a single MLP forward plus per-column softmax, leading to millisecond-level latency; memory scales linearly with active edges.

## 6 LIMITATIONS AND FUTURE WORK

**Dependence on calibration and association.** Geometry features rely on pose/FOV/LOS and timestamp metadata. Systematic yaw drift or time bias can induce coherent but wrong geometry. Our masking and KL–tether reduce harm but are conservative rather than corrective.

**Graph sparsity.** Many items have $|S_n| = 2$, limiting geometric diversity and identifiability; gains over DS grow with degree and saturate at 3+ agents.

**Sensor heterogeneity.** Mixed camera/LiDAR/RSU signals differ in incidence/coverage. The current head encodes agent identity but not modality-specific physics, which can blur cross-modal biases.

**Compute/memory at scale.** Per-edge cost is $O(K)$; storing $\{\Pi_a\}$ is $O(MK^2)$. Dozens of agents and larger $K$ can pressure embedded budgets.

**Interpretability and safety.** While $\Pi_a$ remains interpretable, the context correction $\alpha(x)$ can complicate per-feature attribution in edge cases.

## 7 CONCLUSION

We introduced **PaTSy-Neural-EM**, a DS-compatible extension that injects *state-conditioned* (geometry- and timing-aware) reliability into EM via log-linear corrections on confusion columns. The design preserves DS interpretability, adds gentle trust-region updates for stability, and encodes weak physics priors (range monotonicity, angular smoothness).

Our empirical evidence to date is preliminary: on V2X-Real, PaTSy attains parity top-1 accuracy with DS while yielding improved calibration and stable training under gentle-$\Pi$. Beyond V2X-Real, OPV2V (zero calibration) confirms transfer: our best run improves over DS by +0.9% absolute while remaining stable (Sec. 5.5).

We release implementation details and protocols (see Reproducibility Statement) to facilitate external scrutiny and extensions.

## LLM USAGE STATEMENT

We acknowledge the use of Large Language Models (specifically Claude 4) to assist in the preparation of this manuscript. LLMs were used exclusively for:

- Improving clarity and readability of technical descriptions
- Grammar and style refinement
- Formatting and fixing LaTex Syntax issues
- Ensuring consistency in mathematical notation

  All technical content, experimental design, results, analysis, and scientific conclusions are original work by the authors. The LLM did not contribute to the research methodology, data analysis, or generation of experimental results. We verified all LLM-assisted text for technical accuracy and made corrections where necessary.

- Ensuring consistency in mathematical notation

All technical content, experimental design, results, analysis, and scientific conclusions are original work by the authors. The LLM did not contribute to the research methodology, data analysis, or generation of experimental results. We verified all LLM-assisted text for technical accuracy and made corrections where necessary.

## REPRODUCIBILITY STATEMENT

To preserve double-blind review, we do not include links to code or data at submission time. Upon acceptance, we will release the full artifact as part of the conference process (at camera-ready), including: (i) code for curation (`curate_associate.py`), baselines (`train_baselines.py`), DS/EM (`train_em.py`), Neural-EM (`train_neural_em.py`), and evaluation (`evaluate.py`); (ii) configuration files specifying gating thresholds (range/FOV/$\Delta t$), regularizer weights, and the EM schedule; and (iii) scripts to regenerate all tables and figures from a single JSON log. We fix random seeds, report 5-seed means $\pm$ 95% confidence intervals, and document software/hardware versions (PyTorch 2.x, CUDA 12.x, NVIDIA T4 16 GB).

## A ADDITIONAL TABLES (ALTERNATE EVALUATION PROTOCOL)

## REFERENCES

[1] A. P. Dawid and A. M. Skene. Maximum likelihood estimation of observer error-rates using the EM algorithm. *Journal of the Royal Statistical Society: Series C*, 28(1):20–28, 1979.

Table 4: Cross-method comparison under an alternate protocol (not directly comparable to main V2X-Real tables).

| Method | Train Acc (%) | Val Acc (%) | Test Acc (%) | ECE (VAL) |
|---|---|---|---|---|
| PaTSy-Enhanced (Curriculum) | 93.6 | 97.7 | 91.9 | 0.244 |
| PaTSy-Graph+DS | 94.2 | 97.7 | 91.9 | **0.198** |
| Dawid-Skene (Advanced) | 84.3 | 87.0 | 85.2 | 0.156 |
| GLAD | 78.6 | 78.8 | 79.1 | 0.203 |
| BCC | 79.3 | 79.5 | 79.8 | 0.189 |
| Majority Vote | 78.6 | 78.8 | 79.0 | 0.215 |
| GTM | 75.2 | 0.0 | 75.5 | 0.312 |
| CRH | 76.1 | 76.1 | 76.3 | 0.228 |
| LTM | 25.0 | 25.0 | 25.2 | 0.445 |
| Neural TD | 69.2 | 69.2 | 69.4 | 0.267 |
| Dependency TD | 57.6 | 57.6 | 57.8 | 0.334 |

[2] V. C. Raykar, S. Yu, L. H. Zhao, G. H. V. Valadez, C. Florin, L. Bogoni, and L. Moy. Learning from crowds. *Journal of Machine Learning Research*, 11:1297–1322, 2010.

[3] R. Tanno, A. Saeedi, S. Sankaranarayanan, D. C. Alexander, and N. Silberman. Learning from noisy labels by regularized estimation of annotator confusion. In *CVPR*, 2019.

[4] J. Zhang, X. Wu, and V. S. Sheng. Learning from crowdsourced labeled data: a survey. *Artificial Intelligence Review*, 53(6):4405–4443, 2020.

[5] Q. Chen, S. Tang, Q. Yang, and S. Fu. Cooper: Cooperative perception for connected autonomous vehicles. In *ICDCS*, 2019.

[6] T.-H. Wang, S. Manivasagam, M. Liang, B. Yang, W. Zeng, and R. Urtasun. V2VNet: Vehicle-to-vehicle communication for joint perception and prediction. In *ECCV*, 2020.

[7] Y. Liu, T. Guo, J. Wang, and Y. Wang. When2com: Multi-agent perception via communication graph grouping. In *CVPR*, 2020.

