# OpenReview forum: "PaTSy-Neural-EM: Geometry-Aware Truth Discovery for Real-Time Multi-Agent Perception"
_ICLR.cc/2026/Conference — ICLR 2026 Conference Withdrawn Submission_

### Official Review · Reviewer_pgL5 · 2025-10-28

**Soundness:** 2
**Presentation:** 1
**Contribution:** 2
**Rating:** 0
**Confidence:** 5

**Summary:**

This submission presents PaTSyNeural-EM, a geometry-aware EM framework for multi-agent perception. It provides a new theoretical aspect to study Cooperative V2X systems. The results slightly improves the baselines.

**Strengths:**

1. This submission proposes a new theoretical aspect to study multi-agent perception.
2. The limited results show certain potential of the proposed method.

**Weaknesses:**

1. The submission is not ready for any publication. It is a surprise for me to see such a submission at ICLR.
2. The content is incomplete and very hard to follow. Notably, the related work section has only two lines.
3. Many mathematical symbols are not defined and equations are not well explained at all.

**Questions:**

The lack of clarity makes it challenging to formulate any constructive questions.

---

### Official Review · Reviewer_EDiF · 2025-10-30

**Soundness:** 2
**Presentation:** 2
**Contribution:** 2
**Rating:** 4
**Confidence:** 3

**Summary:**

The paper addresses truth discovery for cooperative perception in V2X scenarios, where multiple agents (vehicles and roadside units) submit labels about the same scene, and the system must infer the most likely ground truth. The key idea is to keep the familiar Dawid–Skene (DS) per‑agent confusion matrices but make them context‑aware: a small neural module adjusts an agent’s reliability on the fly using geometry and timing features such as range, bearing, line‑of‑sight/occlusion proxies, time offset, and agent identity. These adjusted reliabilities are re‑normalized so they remain valid probabilities and are used inside an EM (expectation–maximization) loop. To stabilize training and preserve the interpretability of the DS parameters, the authors use a three‑stage schedule—start from a DS solution, train only the neural head, then fine‑tune everything with a penalty that keeps the solution near the DS initialization—and add physics‑inspired regularizers that encourage reliability to decrease with distance and to vary smoothly with viewing angle. Experiments on V2X‑Real and OPV2V report DS‑level accuracy with better calibration on V2X‑Real and a small accuracy gain on OPV2V, alongside millisecond‑level inference on an NVIDIA T4

**Strengths:**

1. Conceptual simplicity with DS‑compatibility. Casting context‑aware reliability as additive logit corrections to DS confusion columns is clean and keeps the DS interpretation at the core. This makes the approach easier to plug into EM‑style TD pipelines.
2. Physics‑guided priors. Range‑monotonicity and angular‑smoothness regularizers encode domain knowledge and could help prevent pathological fits in sparse regimes

**Weaknesses:**

1. Headline results show no accuracy improvement on the main dataset; calibration evidence is incomplete.
Table 1 (page 3) reports identical validation and test accuracies for DS, Graph+DS, and PaTSy (0.9766 / 0.9190), i.e., no improvement over DS. Meanwhile, Table 2 only reports PaTSy’s ECE (0.244 VAL / 0.299 TEST) without the DS ECE, so readers cannot verify the “improved calibration” claim. Figure 2 (page 4) is explicitly labeled “concept” rather than empirical. This makes the main claim—“better calibration while preserving DS‑level accuracy”—unsubstantiated on the primary benchmark. Please provide actual reliability diagrams and NLL/ECE for all methods.
2. Missing baselines and incomplete related work for context‑aware annotator models.
The proposed idea—context‑ or item‑dependent annotator reliability—has close precedents (e.g., GLAD‑style difficulty modeling, Bayesian Classifier Combination variants, neural DS extensions). While Table 4 lists names like GLAD and BCC under an “alternate protocol,” these are not evaluated in the main setting and are not properly cited in the references section; key works (e.g., GLAD/Whitehill et al.) are missing. The paper should position PaTSy against instance‑dependent/noise‑adaptive TD methods with comparable training budgets on the same splits.

**Questions:**

KL tether anchor. In Eq. (4), is the KL penalty anchored to the DS solution or to the previous iterate? If the former, why call it a trust‑region? Please provide ablations that (i) tether to DS, (ii) tether to previous Π, and (iii) no tether

---

### Official Review · Reviewer_HVxs · 2025-10-31

**Soundness:** 1
**Presentation:** 1
**Contribution:** 1
**Rating:** 0
**Confidence:** 5

**Summary:**

This paper should be desk rejected.

**Strengths:**

This paper should be desk rejected.

**Weaknesses:**

This paper should be desk rejected.

**Questions:**

This paper should be desk rejected.

---

### Official Review · Reviewer_Bouw · 2025-11-02

**Soundness:** 2
**Presentation:** 1
**Contribution:** 2
**Rating:** 2
**Confidence:** 3

**Summary:**

This paper revisits truth discovery for cooperative V2X perception, arguing that classical Dawid–Skene is mis-specified because agent reliability depends on geometry and timing (range, bearing, occlusion, latency, agent identity). The authors propose PaTSy-Neural-EM, which preserves DS interpretability but makes reliability context-conditioned via a lightweight log-linear head and a per-column softmax inside EM; training is stabilized with a gentle-Π schedule (DS warm start → head-only → unfreeze with a KL tether) plus physics-inspired regularizers enforcing range-monotonicity and angular smoothness. On V2X-Real, the method matches DS in top-1 accuracy while improving calibration and robustness on hard slices; on OPV2V under zero calibration, it yields a +0.9% absolute gain, and the authors claim real-time performance.

**Strengths:**

1. Stability recipe that addresses Neural-EM pitfalls: The gentle-Π schedule (DS warm start → head-only → unfreeze with a KL tether) plus physics-inspired regularizers (range-monotonicity, angular smoothness) targets identifiability and training stability in a practical, reproducible manner.
2. Practical signal: On V2X-Real, the method matches DS in top-1 while improving calibration and hard-slice robustness; it also reports real-time feasibility and a small positive transfer on OPV2V under zero calibration.

**Weaknesses:**

1. Insufficient related work coverage. For a 2025 submission, surveying only seven papers (with the most recent around 2020) is inadequate. The literature on collaborative perception contains many more recent efforts that explicitly model confidence or spatial confidence maps and their impact on multi-agent fusion; these should be discussed and, where possible, compared.
2. Ambiguous writing and notation. Several definitions and symbols are introduced with limited exposition, making the derivations and the exact semantics of variables difficult to follow; clearer notation tables and step-by-step derivations would improve readability.
3. Lack of system/dataflow visualization. The paper would benefit from a concise schematic of the overall setting and per-item dataflow (agents, gating, features, EM steps). Current figures focus on accuracy/calibration/dynamics but do not convey the end-to-end pipeline.
4. Underspecified experimental setting and limited datasets. Important protocol details are condensed, and the evaluation is restricted to V2X-Real and OPV2V, limiting external validity; broader datasets and fuller ablations/baselines are needed to substantiate generality.

**Questions:**

1. EM details & stability. Specify the exact objective optimized per phase (DS-warm start / head-only / unfreeze with KL), the update rules, number of inner E-steps, stopping criteria, and evidence of monotonic improvement. Ablate the gentle-Π schedule and the KL tether to show their necessity.
2. Baselines & related work coverage. Expand the survey and comparisons to include stronger TD/Neural-EM baselines (e.g., GLAD, MACE, Bayesian DS, recent neural variants) and recent collaborative-perception methods that model confidence (confidence maps, spatial gating, flow-based reuse). Justify any omissions.
3. Datasets, protocol, and generalization. Clearly detail splits/protocols and broaden evaluation (e.g., DAIR-V2X, V2X-Sim). Define the “zero-calibration” setting precisely. Visualize conditional confusion columns across agents and report efficiency scaling with the number of agents/classes.

---

### Note · Authors · 2025-11-12

I have read and agree with the venue's withdrawal policy on behalf of myself and my co-authors.